# Surface Defect Detection of Hot Rolled Steel Based on Attention Mechanism and Dilated Convolution for Industrial Robots

Yuanfan Yu [1], Sixian Chan [1,2,*], Tinglong Tang [2], Xiaolong Zhou [3], Yuan Yao [4] and Hongkai Zhang [5]

[1] College of Computer Science and Technology, Zhejiang University of Technology, Hangzhou 310023, China
[2] Hubei Key Laboratory of Intelligent Vision Based Monitoring for Hydroelectric Engineering, The College of Computer and Information, China Three Gorges University, Yichang 443002, China
[3] The College of Electrical and Information Engineering, Quzhou University, Quzhou 324000, China
[4] The School of Computer Science, University of Nottingham Ningbo, Ningbo 315100, China
[5] The School of Electronic and Information Engineering, Anhui Jianzhu University, Hefei 230022, China
* Correspondence: sxchan@zjut.edu.cn; Tel.: +86-173-5722-8908

**Abstract:** In the manufacturing process of industrial robots, the defect detection of raw materials includes two types of tasks, which makes the defect detection guarantee its accuracy. It also makes the defect detection task challenging in practical work. In analyzing the disadvantages of the existing defect detection task methods, such as low precision and low generalization ability, a detection method on the basis of attention mechanism and dilated convolution module is proposed. In order to effectively extract features, a two-stage detection framework is chosen by applying Resnet50 as the pre-training network of our model. With this foundation, the attention mechanism and dilated convolution are utilized. With the attention mechanism, the network can focus on the features of effective regions and suppress the invalid regions during detection. With dilated convolution, the receptive field of the model can be increased without increasing the calculation amount of the model. As a result, it can achieve a larger receptive field, which will obtain more dense data and improve the detection effect of small target defects. Finally, great experiments are conducted on the NEU-DET dataset. Compared with the baseline network, the proposed method in this paper achieves 81.79% mAP, which are better results.

**Keywords:** deep learning; defect detection; attention mechanism; dilated convolution





## 1. Introduction

When hot rolled steel is forged, it needs a higher ambient temperature. Although its strength is not very good, its plasticity and weldability are good enough to satisfy the demands of people's daily life, as well as industrial robot manufacturing, such as hot rolled steel coil marking robots. Hot rolled products are widely applied in shipbuilding, automobile, bridge, construction, machinery, pressure vessel, and other manufacturing industries because of their high strength, good toughness, easy processing, and benign weldability. However, due to the complexity of the environment or the mistakes of the operation, various defects are often inevitable during the manufacturing processes of hot rolled steel. If some defects are not detected, it may lead to serious industrial accidents, and even cause casualties. There are six kinds of common steel defects in industry, including patches (Pa), crazing (Cr), pitted surface (PS), inclusion (In), rolled-in scale (RS) and scratches (Sc). In many actual industrial scenes, most of the defect types have been professionally counted and summarized in advance, so the defect characteristics can be used for direct detection in method design, or prior knowledge and labeled data sets can be used to train the model. In traditional ideas, industrial detection is usually through the color, shape and other characteristics of defects in the image, using image processing methods and combining traditional machine learning to detect. From the view of an image,

since industrial defects are often emanated from sudden variations of pixels in the image, the edge detection method can quickly and effectively find the defect area. The commonly used edge detection operators include Prewitt, Sobel and Canny. However, those methods are hard to handle the complex background or low SNR defects and require high imaging conditions. From the perspective of the frequency domain, abrupt defects often show high-frequency characteristics in the frequency spectrum. Therefore, for products with simple or periodic backgrounds, Fourier transform (FT) [1], Gabor transform, wavelet transform and other methods can be used to transform to the frequency domain for detection. Chetverikov et al. [2] detect abrupt defects based on the texture direction of textile surface. Hou et al. [3] use Gabor wavelet transform operators suitable for texture representation to extract the frequency domain information of images, and then use the support vector machine (SVM) to distinguish different types. However, the interior of regional industrial defects is usually relatively stable, so the methods based on edge detection can usually only detect the defect boundary. However, large-area defects will affect the statistical features of the image, which can be characterized by such statistical methods as the difference of gray level changes [4], gray level histogram [5], and color features. Based on the above defect descriptors, machine learning methods (such as random forest [6] and SVM) can also be further applied for classification and teaching [7].

With the development of the theory of deep learning, image processing methods based on CNN [8] are booming. CNN has a strong ability of feature extraction, which can be used not only for image processing but also for natural language processing, such as automatically displaying natural language sentences of images through computers [9] and robot path planning [10]. The CNN-based methods have gradually become the mainstream to extract features for classification [11]. Cha et al. [12] detect bridge defects based on the two-stage detector Faster R-CNN [13] and change the backbone network to ZF-net to improve real-time performance. Tao et al. [14] cascade Faster R-CNN to realize defect detection of insulators in power patrol inspection. He et al. [15] improve detection accuracy by fusing multi-scale features. Zhang et al. [16] use YOLOv3 [17] to detect bridge defects and use migration learning, focal loss and batch re-standardization to raise the detection efficiency. Chen et al. [18] combine DenseNet [19] with YOLOv3 to detect LED defects. Yu et al. [20] propose a lightweight network WM-PeleeNet based on the PeleeNet module to identify defect patterns in wafer maps. Moreover, YOLOv5 is used for rotating detection [21]. Plainly, the adjustment of those methods mainly lies in two ideas: introducing a lightweight network to raise the speed of detection, and improving the accuracy of detection by using multi-scale fusion and data enhancement.

In the manufacturing process of industrial robots, the defect detection of raw materials includes two types of tasks, which makes the defect detection guarantee its accuracy. It also makes the defect detection task challenging in practical work. In analyzing the disadvantages of the existing defect detection task methods, such as low precision and low generalization ability, we propose a detection method based on an attention mechanism and dilated convolution module for the surface defect detection of hot rolled steel. To effectively extract features, a two-stage detection model is built as the baseline by using Resnet50 as pre-training network construction. The contributions are as follows:

(1) The attention mechanism is designed to enable our model for focusing on the features of effective areas and suppress invalid areas to obtain robust features.

(2) Dilated convolution is applied to increase the receptive field without increasing the amount of calculation. It has a larger receptive field, which can obtain more dense data and raise the detection effect of small target defects.

## 2. Related Work

Defect detection gained attention in the 1980s. It has been a hot topic in the world, thus contributing to the significant, rich results of relevant research. Those results can be further summarized as filtering methods, statistical methods, and model methods. Since 2013, deep learning technology represented by convolutional neural networks (CNNs) has

successfully been applied and obtained excellent results in a lot of computer vision tasks. The researchers have applied deep learning methods for the defect detection of objects on the surface of the object.

### 2.1. Statistical Method

Tsai et al. [22] established a weighted variance matrix in the neighborhood of pixels and identified defective pixels according to the difference between the two eigenvalues of the variance matrix. Wang et al. [23] designed a new method for feature extraction based on LBP. Kittler et al. [24] applied the K-means for clustering the surface texture into different binary clusters and used the blob method to analyze and detect the defects of ceramic tile surface texture. Wang et al. [25] fused the GLCM and HOG to represent the global and local features of surface texture respectively, and the classification accuracy of five types of steel surface defects reaches 91% mAP.

### 2.2. Spectrographic Method

Tsai et al. [26,27] advance a new method based on FT for the detection of textiles and wood. Chen et al. [28] adopt image analysis methods similar to [29] but select lower-dimensional frequency domain features to identify four main types of defects on the fabric surface. Choi et al. [30] proposed a multi-filter combination method. On the basis of Gabor filter features, ref. [31] introduced GLCM features, morphological features [30], fractal dimension features [31], and other statistical features, effectively improving the detection accuracy. Jasper et al. [32] designed an adaptive wavelet method to achieve new basis functions via the detected texture scenes. Kumar and Gupta [33] used the Hal wavelet to analyze surface texture images and identified defects based on the mean and variance of wavelet function coefficients.

### 2.3. Model Method

These methods analyzed texture properties, built texture image representations, and then detected defects by identifying anomalous textures. Models applied to surface defect detection include the autoregression model [34], Markov random field (MRF) model [35] and texture example (TEXEM) model [36]. Cohen et al. [35] use GMRF to obtain the normal texture. Based on the GMRF model, the issue of defect detection is transformed into a hypothetical detection problem in statistics. Based on this algorithm, ref. [37] further implement a real-time defect detection system that can be used in the industrial field. Xie et al. [36] propose a TEXEM model that can be applied to detect random textures. In this model, texture features are regarded as the superposition of a series of TEXEMs. Each TEXEM can be represented by the mean and variance, and the Gaussian mixture can be used to meet this representation. The TEXEM model is superior to the Gabor filter in the detection of ceramic tile surface defects. Xu et al. [38] use the wavelet domain hidden Markov tree (HMT) to segment the strip surface image. HMT utilizes the Gaussian mixture model to build the distribution of wavelet coefficients in the same scale and uses the quadtree of wavelet coefficients to model the coefficient correlation between scales.

### 2.4. Deep Learning Method

In deep learning-based methods, features are automatically extracted from labeled data, which refrains from the difficulty of manually designing features. At the same time, those methods have a strong ability to classify similar targets. AlexNet [39] was the first convolutional neural network used for large-scale image classification tasks [40]. Shang et al. [41] put forward a method to identify rail defects. Akram et al. [42] design VGG-8, VGG-7 and VGG-6 structures based on VGG-11 network structure for detecting surface defects of solar cells. Faster R-CNN [43] received high attention because of its robustness and efficiency and had also been widely applied in industrial fields. For example, in the field of civil construction [13], the damage detection of cracks in concrete and steel reinforcement was much higher than traditional methods. One-stage detection networks,

such as YOLO [12], single-shot multibox detector (SSD) [44] and CornerNet [45], do not need to extract candidate regions and directly calculate the location and category of objects on the output layer, achieving higher real-time performance. Deng et al. [46] use the YOLO2 network to detect crack defects on concrete surfaces. By adding graffiti interference to the collected images, the training network detects real defects in the complex background and interference.

## 3. Proposed Method

This section mainly introduces the main processes of the network. Figure 1 illustrates the framework of the proposed method. As mentioned above, the Faster R-CNN is selected as the baseline in our experiment.

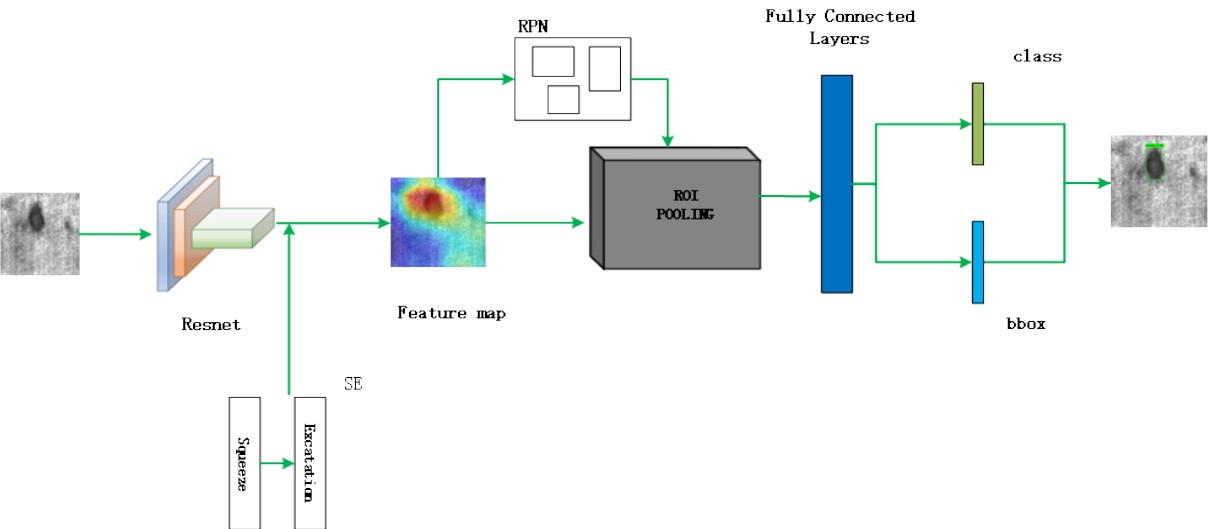

**Figure 1.** Overall framework of the proposed method.

### 3.1. Attention Mechanism

The attention mechanism facilitates the model to pay attention to the region of interest and suppresses irrelevant information. The results are usually displayed in the form of a probability graph or feature vector. When people look at a picture, they tend to focus on the details of some specific areas first. They will ignore the areas where they are not interested. For example, in industrial defect detection, the network will pay attention to the defect area, and the image background information may be purposefully ignored by the model. The overall flow of attention mechanisms is shown in Figure 2. $W$ and $H$ indicate the width and height of the feature map, respectively. $C$ is the number of channels. The size of the input feature map is $W \times H \times C$.

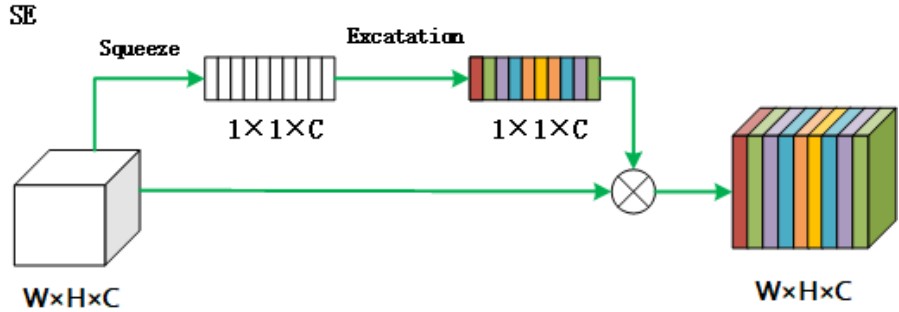

**Figure 2.** The overall flow of attention mechanisms.

The SE attention mechanism [47] first compresses the two-dimensional feature (*W* \* *H*) of each channel into a real number through the global average pooling, and compresses the feature graph from [*w*, *h*, *c*] to [1, 1, *c*], with the following formula:

$$z_c = F_{sq}(u_c) = \frac{1}{H * W} \sum_{i=1}^{H} \sum_{j=1}^{W} u_c(i, j) \tag{1}$$

where $z_c$ is the result of pooling. The embedded position of SE in Conv block and the identification block is shown in Figure 3.

Then, the weight value for each channel is generated, and the correlation between channels is built through two fully connected layers. The number of output weight values is equal to the number of channels. Finally, the previously obtained weights are weighted according to the characteristics of each channel, and the weight coefficient is multiplied by each channel through multiplication. In order to improve the comprehensive performance of the model, cross-channel interaction is relatively used. The core idea of this mechanism is to automatically learn the feature weight according to the value of the loss function through the fully connected network, rather than judging directly via the numerical distribution of the feature channel. Hence, the weight of the effective feature channel can be increased, and the model can be more accurate by adding a small number of parameters and computations. This amplification has more nonlinearity, which can better fit the complex associations between channels.

The introduction of the SE attention module can enhance the characteristics of important channels and weaken the characteristics of unimportant channels. To some extent, the noise interference of the image is eliminated, as well as the accuracy of defect detection being improved.

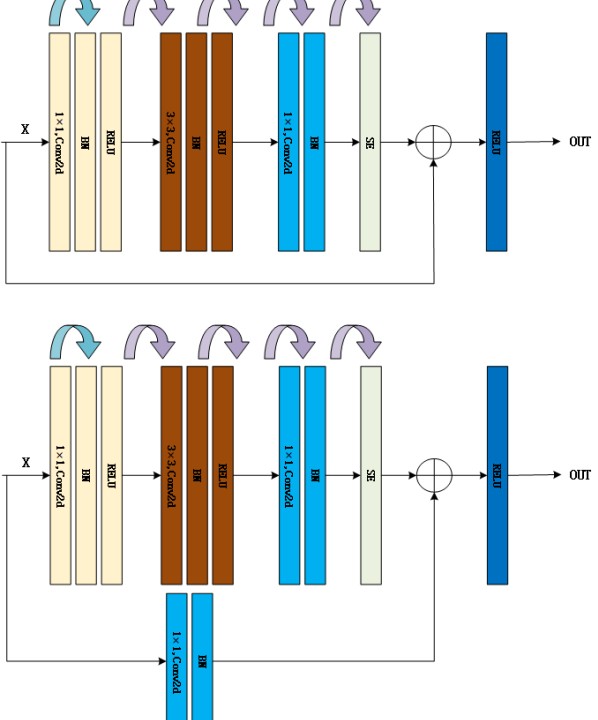

**Figure 3.** Convolution block without/with attention mechanism (up/down).

### 3.2. Dilated Convolution

With the downsampling of images, semantic information will continue to increase, while the resolution will continue to decrease at the same time. Therefore, the high-level feature map contains low-resolution but rich semantics information and is only a small

part of the original picture. However, the size of shallow semantic features varies greatly. After the backbone features are extracted and upsampled, the single-scale feature of the network output can only match certain defects, which will lead to poor defect detection. Therefore, to enhance the semantic extraction of features, expanding the receptive field and multi-scale feature pyramids are becoming commonly used means. Even if this method can extract better shallow semantics, feature extraction for industrial defect detection will destroy high-level features.

Dilated convolution can expand the receptive field without increasing the sum of computation since it uses sparse kernels for convolution. When the expansion rate is 1 and the size of the convolution kernel sets as 3, the operation is the same as the normal convolution. However, if the expansion rate is set as 3, the size of the convolution kernel will be expanded to 7, and the blank part will be filled with 0. In this way, the convolution kernel originally $3 \times 3$ will be expanded to $7 \times 7$. Therefore, the dilation convolution can expand the receptive field but does not bring the amount of computation, thus improving the detection accuracy. The detailed schematic diagram is shown in Figure 4. With the above analysis, an enhanced model of expansion features embedded in dilated convolution is proposed. As shown in Figure 5, it is placed after the Resnet.

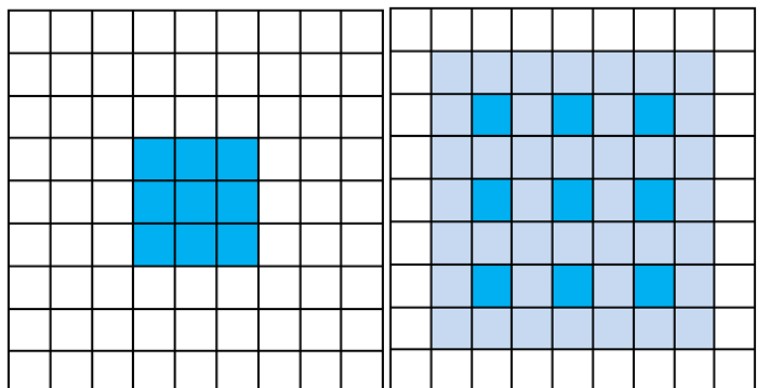

**Figure 4.** The dilation rate on the left is 1 and its receptive field is the same as the conventional $3 \times 3$ convolution. The dilation rate on the right is 3 and its feeling receptive field expands to $7 \times 7$.

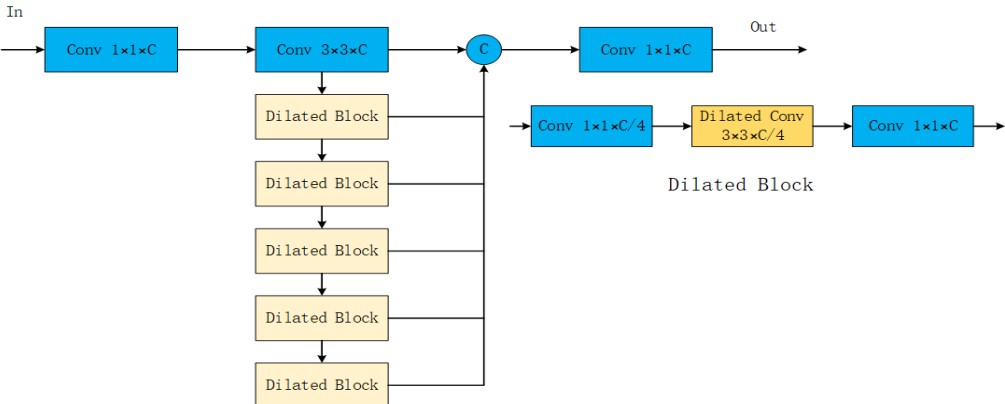

**Figure 5.** The architecture of dilated convolution. Convolution ($k \times k \times C$) means that the number of channels is $C$ and the size of the convolution kernel is $k \times k$.

### 3.3. Loss Function

Defect detection in our work is a supervised learning process. In order to obtain an accurate model, more data are needed for training. For the defect detection task, there are two problems that need to be solved. The first one is defect location, and the other is defect classification. The label of the defect location information and the result are

obtained through testing. If the two stages can completely overlap, the prediction result is 100% correct. However, this is almost impossible in the defect detection task. Because the prediction stage is actually a regression problem, it is difficult for the model to have a completely correct prediction box when dealing with the regression problem. In this paper, we propose a two-stage joint loss function as the classification of prediction boundary boxes. The principle of loss functions is described below. As shown in the following formula, the classified loss and the regression loss together constitute the joint loss function. To adjust the regression and classification task, a regression loss weight $\lambda$ is designed and set as 2 in our experiment:

$$L(\{p_i\}, \{t_i\}) = \frac{1}{N_{cls}} \sum_i L_{cls}(p_i, p_i^*) + \lambda \frac{1}{N_{\mathrm{reg}}} \sum_i p_i^* L_{\mathrm{reg}}(t_i, t_i^*) \tag{2}$$

For classification loss, $N_{cls}$ represents the size of batchsize. $p_i^*$ is the ground-truth label. If the anchor is a positive sample, $p_i^*$ is 1; otherwise, $p_i^*$ is 0. $p_i$ is the prediction probability of the target. Classification loss $L_{cls}$ belongs to cross-entropy loss, and the formula is as follows:

$$L_{cls}(p_i, p_i^*) = -\log[p_i p_i^* + (1 - p_i^*)(1 - p_i)] \tag{3}$$

For regression loss, $N_{\mathrm{reg}}$ is the number of anchor boxes. $t_i = \{t_x, t_y, t_w, t_h\}$ is a quadruple that indicates the four parameter coordinates of the predicted boundary frame. $t_i^*$ is the coordinate vector of the ground-truth box.

$$L_{\mathrm{reg}}(t_i, t_i^*) = R(t_i - t_i^*) \tag{4}$$

where $R$ is the smooth $L1$ function.

$$R(\mathrm{x}) = \begin{cases} 0.5x^2 & (|x| < 1) \\ |x| - 0.5 & \text{otherwise} \end{cases} \tag{5}$$

where $x$ is $t_i - t_i^*$.

In addition, adding $L2$ regularization to the loss function can prevent the possibility of overfitting the model during training. This formula is shown below:

$$L_{total} = L(\{p_i\}, \{t_i\}) + \frac{\gamma}{2n} \sum_w w^2 \tag{6}$$

Among them, the number of samples contained in the training set is represented by $n$. The weight parameter during training is represented by $w$, and $\gamma$ is the weight attenuation coefficient.

## 4. Experiment

### 4.1. Dataset

In the current experiment, we chose an open source data set called NEU-DET [14]. This data set contains six different types of defects, including rolled-in scale (RS), patches (Pa), crazing (Cr), pitted surface (PS), inclusion (In) and scratches (Sc). The data set has a total of 1800 images with a size of $200 \times 200$, and each type of defect has 300 images.

From Figure 6, it is evident that differences can be found between different types of defects, such as patches, crazing, pitted surface, etc. There are also discrepancies within the same category of defects, such as scratch defects, which include horizontal scratches, vertical scratches, inclined scratches, etc. In addition, images with the same defect category may also have different gray levels because of variations in light and material.

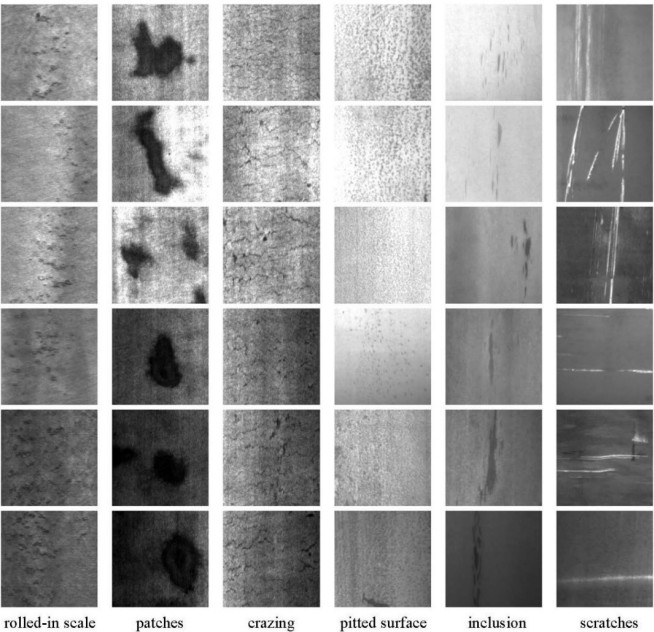

**Figure 6.** The samples of steel surface defect from the NEU-DET date set, including rolled-in scale, patches, crazing, pitted surface, inclusion and scratches, respectively (from left to right).

### 4.2. Evaluation Metrics

Different from the task of defect classification, defect detection cannot only take accuracy as an index for evaluating the performance of the entire network. Since $mAP$ is an indicator that can combine accuracy and recall, we use it to evaluate the comprehensive performance of the proposed model in experiments:

$$precision = \frac{TP}{TP + FP} \tag{7}$$

$$recall = \frac{TP}{TP + FN} \tag{8}$$

$$mAP = \frac{1}{|N|} \sum_{i \in N} AP(i) \tag{9}$$

where $TP$, $FP$, $FN$ and $TN$ represent positive samples with correct prediction, positive samples with wrong prediction, negative samples with wrong prediction and positive samples with correct prediction, respectively. *Precision* means the probability of the correct prediction of positive samples. *Recall* represents the probability of regarding the correct prediction as a positive sample in the positive sample of the original sample. $AP$ represents the sum of precision on each image and the total number of images containing this category, and $mAP$ is the average $AP$ value of each category.

### 4.3. Setting and Training

In our experiments, the original data set is randomly split into a training set (including training data and verification data) and a test set in the form of 9:1. Then, the training set and verification set are randomly separated in the form of 9:1. That is to say, the 1800 images in the whole dataset are divided into 1458 for training, 162 for verification and 180 for testing. For the hardware environment, the experiments are run on a server with two 2080Ti. For training, stochastic gradient descent (SGD) is chosen as the optimizer, while setting a variable learning rate and optimizing the learning rate via the cosine annealing method. Notably, the backbone network is frozen during the training process to reduce the problems caused by machine performance. The epoch is set as 200. In addition, we

use mosaic and mixup methods to enhance training data so as to obtain the parameters of the model and ameliorate its generalization capability. Finally, we conduct experiments about the effects of the pretraining network Resnet on several networks. Furthermore, the effects of various attention mechanisms on this model are also analyzed. We set the initial value of the learning rate to $1 \times 10^{-4}$, set the minimum learning rate to $1/100$ of the initial value, set the momentum to 0.9, and set the learning rate attenuation strategy to cos decrease. For training parameters: we set the batch size to 8, and the optimizer selects "Adam". At the same time, we use the method of freezing the trunk to conduct training, freezing the first 100 epochs.

### 4.4. Comparison with the-Sate-of-Art Methods on NEU-DET

In order to further validate the effectiveness of the proposed model, we conducted experiments on NEU-DET to compare with the-state-of-the-art methods, including Faster R-CNN [48], CenterNet [49], RetinaNet [50], YOLOv4 [51], YOLOv4 [51], YOLOv7 [52] and Cascade R-CNN [53]. Due to the accidental consideration of experimental data, the data shown in the table are the average values of the map after 10 runs. Table 1 shows the defect detection experiments of different models. It can be seen that our method reaches 81.35% ($mAP$), increased by 9.41% (compare to YOLOv4) and 6.52% (compare to YOLOv7). This shows that at present, the performance of the two-stage detection model is better than the one-stage model in our experiments.

**Table 1.** Defect detection experiments results of different models on NEU-DET.

| Network Model | Backbone | mAP | Model Size |
|---|---|---|---|
| Baseline | Resnet50 | 77.72 | 124 M |
| Faster R-CNN [48] | VGG16 | 66.14 | 150 M |
| CenterNet [49] | Resnet50 | 70.07 | 148 M |
| RetinaNet [50] | Resnet50 | 68.54 | 147 M |
| YOLOv4 [51] | CSPDarkNet53 | 71.94 | 224 M |
| YOLOv7 [52] | E-ELAN | 74.83 | 144 M |
| Cascade R-CNN [53] | Resnet50 | 73.33 | 210 M |
| Ours | Resnet50 | 81.79 | 114 M |

### 4.5. Ablation Study

We noticed that in the original model's detection, the model is too focused on local small features, and we can change the model's focus on target defects by adding dilated convolution and attention mechanisms. Based on the baseline, two components (SE and dilation) are designed in our model. SE stands for adding an attention mechanism to the original network, and dilation stands for adding a dilated convolution module to the network. In order to quantitatively analyze the performance of each component in the proposed method, we conduct ablation experiments. Table 2 shows the experiment's results of the $mAP$ value of the model with different components, respectively. The two components improve the $mAP$ by 1.77% (SE) and 2.33% (dilation), respectively. The $mAP$ value reaches 81.79% when both components are added, which increases by 4.07% than the baseline. The experiment results illustrate that both components are effective in our model.

**Table 2.** Performance of different components on the NEU-DET.

| | CR | RS | SC | IN | PA | PI | mAP |
|---|---|---|---|---|---|---|---|
| Baseline | 0.41 | 0.75 | 0.96 | 0.84 | 0.96 | 0.75 | 77.72 |
| Baseline + SE | 0.55 | 0.71 | 0.91 | 0.84 | 0.98 | 0.79 | 79.49 |
| Baseline + dilation | 0.50 | 0.75 | 0.95 | 0.85 | 0.97 | 0.77 | 80.05 |
| Baseline + dilation + SE | 0.59 | 0.71 | 0.96 | 0.85 | 0.98 | 0.83 | 81.79 |

*4.6. Analysis of Attention Mechanisms*

To further demonstrate the influence of the attention mechanism in the proposed method, we conduct analysis studies by comparing with the baseline and ECA [54] attention on NEU-DET. Table 3 shows the effect of the attention mechanism on the experimental results. It can be seen that the experimental results are improved by 1.77% and 1.56% after using the SE attention mechanism and the ECA attention mechanism, respectively, compared to the baseline. This indicates that the channel attention mechanism can improve the accuracy of the model. Once the network is fortified with the attention mechanism, it can focus on the region that is more attractive, while ignoring the invalid region or even the area with a large noise impact. As a result, the application of SE attention can promote the detection ability of the model.

**Table 3.** Influence of different attention mechanisms on NEU-DET.

| Attention | mAP |
| --- | --- |
| Baseline | 77.72 |
| Baseline + SE | 79.49 |
| Baseline + ECA | 79.28 |

*4.7. Visualization*

As shown in Figure 7, the visualization of detection results can be seen from top to bottom, and from left to right, a display of the six types (crazing (Cr), inclusion (In), patches (Pa), pitted surface (PS), rolled-in scale (RS) and scratches (Sc)) in order, with the confidence level of the corresponding class above the box.

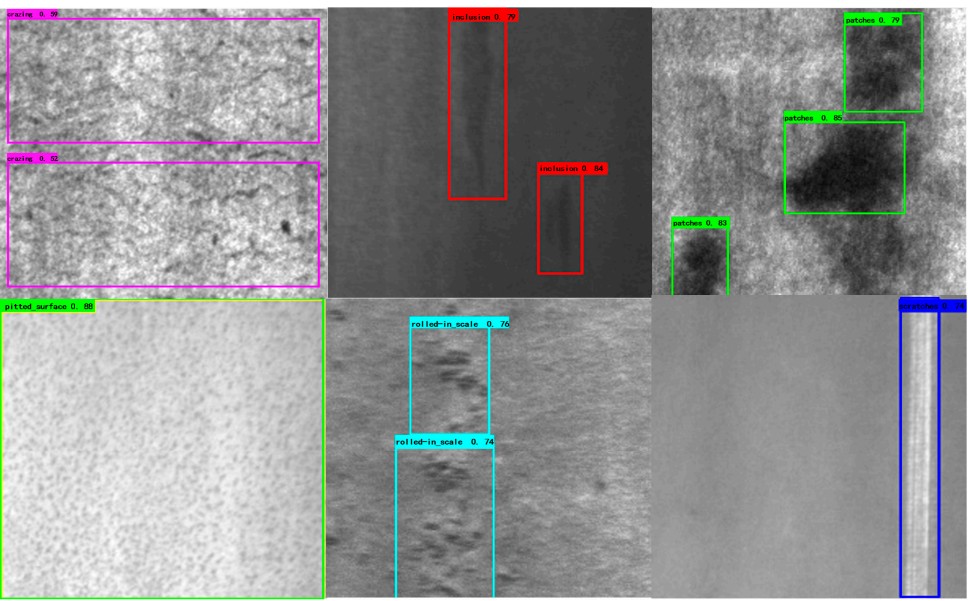

**Figure 7.** Visualization of detection results.

**5. Conclusions**

In this paper, we proposed a new approach based on the attention mechanism and dilated convolution for detecting the surface defect detection of hot rolling steel in the robot system. In order to effectively extract features, Resnet50 was used as the pre-training backbone, and a two-phase detection model was constructed as the baseline network. The attention mechanism was designed to enable the network for focusing on the features of effective areas and suppress invalid areas. The application of dilated convolution enhancement could expand the receptive field without bringing any additional calculation, and effectively improve the detection accuracy. The proposed method in this paper

achieved 81.79%, which achieved a better result. The experimental results illustrated that the performance of the network in the surface defect detection of hot rolled steel was better than the current mainstream target detection network model. However, for certain categories, defects cannot be detected well, which will have a significant impact on the production of steel. In the future, we will further optimize the model.

**Author Contributions:** Conceptualization, Y.Y. (Yuanfan Yu) and S.C.; investigation, Y.Y. (Yuanfan Yu) and H.Z.; methodology, Y.Y. (Yuan Yao), S.C. and T.T.; software and validation, Y.Y. (Yuanfan Yu); writing—original draft preparation, Y.Y. (Yuanfan Yu); writing—review and editing and funding acquisition, S.C. and X.Z. All authors have read and agreed to the published version of the manuscript.

**Funding:** This work was supported by the National Natural Science Foundation of China (Grant No. U20A20196, 61906168 and 62272267); Zhejiang Provincial Natural Science Foundation of China (Grant No. LY23F020023); Joint Funds of the Zhejiang Provincial Natural Science Foundation of China (Grant No. LZJWZ22E090001), and Construction of Hubei Provincial Key Laboratory for Intelligent Visual Monitoring of Hydropower Projects (Grant No. 2022SDSJ01), the Hangzhou AI major scientific and technological innovation project (Grant No. 2022AIZD0061) and Yongjiang Talent Introduction Programme (Grant No. 2022A-234-G).

**Institutional Review Board Statement:** Not applicable.

**Informed Consent Statement:** Not applicable.

**Data Availability Statement:** This study did not report any data. We used public data for research.

**Conflicts of Interest:** The authors declare no conflict of interest.

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
