# Peer review of "Surface Defect Detection of Hot Rolled Steel Based on Attention Mechanism and Dilated Convolution for Industrial Robots"

_electronics, doi:10.3390/electronics12081856_

Round 1

Reviewer 1 Report

This paper presented a novel technique that employs dilated convolution and attention mechanisms to identify surface flaws in hot-rolled steel within a robotic system. The methodology and the experiment results are interesting.  The authors should extend the conclusions with future work for this article.

1- The symbols in equation no 9 should be defined further.

2- The authors should explain why they choose (SDG) as optimizer for training the models.

3- Common issue when training a model is overfitting. How you will address the overfitting problems.

4- There are many activation functions mentioned in the manuscript such as fully connected layer, pooling layers , and convolutional layers , justify the usage of these layers?

5- The authors should extend the conclusions with future work for this article.

Reviewer 2 Report

In the manufacturing process of industrial robots, the defect detection of raw materials includes two types of tasks, which makes the defect detection guarantees its accuracy. And it also makes the defect detection task challenging in practical work. In analyzing the disadvantages of  the existing defect detection task methods, such as low precision and low generalization ability, a detection method on the basis of attention mechanism and dilated convolution module is proposed. In order to effectively extract features, a two-stage detection framework is chosen by applying Resnet50 as the pre-training network of proposed model. With this foundation, the attention mechanism and dilated convolution are utilized. With the attention mechanism, the network can focus on the features of  effective regions and suppress the invalid regions during detection. With dilated convolution, the  receptive field of the model can be increased without increasing the calculation amount of the model. As a result, it can achieve a larger receptive field, which will obtain more dense data and improve the detection effect of small target defects. Finally, great experiments are conducted on the NEU-DET dataset. Compared with the baseline network, the proposed method in this paper achieves 81.79 % 13 mAP, which gets better results. Author must address my following queries:

1.    The proposed research is evaluate while using NEU-DET dataset, authors must justify that why did they used only 1 dataset for proposed research evaluation.

2.    The parameter optimization details are missing

3.    There are 2 claims as contributions of this work, how each claim has been obtained and proposed, a detail is required with each claim that the experiment A prove this and experiment b prove this etc

4.    The comparison with classical machine learning algorithms is required to be presented.

5.    Authors are suggested to cite and discuss

‘De Luca, Gennaro, and Yinong Chen. "Explainable artificial intelligence for workflow verification in visual IoT/robotics programming language environment." Journal of Artificial Intelligence and Technology 1, no. 1 (2021): 21-27.’

‘Zan, J., 2022. Research on robot path perception and optimization technology based on whale optimization algorithm. Journal of Computational and Cognitive Engineering, 1(4), pp.201-208.’

‘Yu, Naigong, Huaisheng Chen, Qiao Xu, Mohammad Mehedi Hasan, and Ouattara Sie. "Wafer map defect patterns classification based on a lightweight network and data augmentation." CAAI Transactions on Intelligence Technology (2022).’

6.    Manuscript requires corrections of English/grammar

7.    The limitations of proposed research must be discussed

8.    Values presented in table and graphs must be statistically verified

Reviewer 3 Report

Dear authors, 

Congratulations on completing your research successfully. However, I have feedback for your team: You have mentioned that with dilated convolution, the receptive field of the model can be increased without increasing the calculation of the model so I think it will be better if you can show it in the result such as memory or the number of parameters of your model as compared with others.

Additionally, I suggest you should replace the phrase 'full connected layer' with 'fully connected layer'. This is a tiny grammar issue you can consider checking.

Sincerely.

Round 2

Reviewer 2 Report

Authors have addressed my comments and i am recommending acceptance of this paper.

Reviewer 3 Report

Dear authors,

I have received your response to my reviews but it is not what I mean.  I want you to show some results (maybe an additional column in your table) to compare the number of 'parameters' i.e. the number of weights and bias, not the 'hyperparameters' you mentioned. It can help readers easy to access your model and see its outstanding.

Thanks.
